


# Investigating Diesel Engines as an Atmospheric Source of Isocyanic Acid in Urban Areas

Shantanu H. Jathar[1], Christopher Heppding[1], Michael F. Link[2], Delphine K. Farmer[2], Ali Akherati[1], Michael J. Kleeman[3], Joost A. de Gouw[4, 5], Patrick R. Veres[4, 5], and James M. Roberts[4]

[1] Department of Mechanical Engineering, Colorado State University, Fort Collins, CO, USA, 80523
[2] Department of Chemistry, Colorado State University, Fort Collins, CO, USA, 80523
[3] Department of Civil and Environmental Engineering, University of California Davis, Davis, CA, USA, 95616
[4] NOAA Earth System Research Laboratory, Chemical Sciences Division, Boulder, CO, USA, 80305
[5] Cooperative Institute for Research in Environmental Sciences, University of Colorado, Boulder, CO, USA, 80305

*Correspondence to*: Shantanu H. Jathar (shantanu.jathar@colostate.edu)

**Abstract.** Isocyanic acid (HNCO), an acidic gas found in tobacco smoke, urban environments and biomass burning-affected regions, has been linked to adverse health outcomes. Gasoline- and diesel-powered engines and biomass burning are known to emit HNCO and hypothesized to emit precursors such as amides that can photochemically react to produce HNCO in the atmosphere. Increasingly, diesel engines in developed countries like the United States are required to use Selective Catalytic

Reduction (SCR) systems to reduce tailpipe emissions of oxides of nitrogen. SCR chemistry is known to produce HNCO as an intermediate product, and SCR systems have been implicated as a atmospheric source of HNCO. In this work, we measure HNCO emissions from an SCR system-equipped diesel engine and, in combination with earlier data, use a three-dimensional chemical transport model (CTM) to simulate the ambient concentrations and source/pathway contributions to HNCO in an urban environment. Engine tests were conducted at three different engine loads, using two different fuels and at

multiple operating points. HNCO was measured using an acetate chemical ionization mass spectrometer. The diesel engine was found to emit primary HNCO (3-90 mg kg-fuel$^{-1}$) but we did not find any evidence that the SCR system or other aftertreatment devices (i.e., oxidation catalyst and particle filter) produced or enhanced HNCO emissions. The CTM predictions compared well with the only available observational data sets for HNCO in urban areas but under-predicted the contribution from secondary processes. The comparison implied that diesel-powered engines were the largest source of

HNCO in urban areas. The CTM also predicted that daily-averaged concentrations of HNCO reached a maximum of ~110 pptv but were an order of magnitude lower than the 1 ppbv level that could be associated with physiological effects in humans. Precursor contributions from other combustion sources (gasoline and biomass burning) and wintertime conditions could enhance HNCO concentrations but need to be explored in future work.

## 1 Introduction

Isocyanic acid (HNCO) is a mildly acidic gas, which is highly soluble at physiologic pH and can participate in carbamylation reactions in the human body (Wang et al., 2007) and lead to adverse health outcomes such as cataracts, atherosclerosis, and rheumatoid arthritis (Fullerton et al., 2008;Scott et al., 2010). Isocyanates, the family to which HNCO belongs, are extremely hazardous. The accidental release of methyl isocyanate from a pesticide plant in Bhopal, India in 1984 resulted in thousands of deaths and hundreds of thousands in injuries within weeks of the release (Broughton, 2005).

Although isocyanates are understood to be toxic and regulated through best practices in indoor and occupational environments (Alexeeff et al., 2000;SWEA, 2005), it is unclear if ambient concentrations of HNCO (and isocyanates in general) are high enough for it be of concern as an outdoor air pollutant. Roberts et al. (2011) proposed that exposure to HNCO concentrations exceeding 1 ppbv could be harmful to humans.





Only a few studies have measured ambient concentrations of HNCO. Roberts and coworkers (Roberts et al., 2011;Roberts et al., 2014) used a chemical ionization mass spectrometer (CIMS) to measure HNCO in urban areas (Pasadena, CA) and biomass burning-affected regions (suburban and rural Colorado). While concentrations in urban areas were consistently below 100 pptv, ambient HNCO concentrations exceeded 100 pptv in regions affected by wildfire plumes and agricultural

burning; in case of the later source, HNCO concentrations reached as high as 1.2 ppbv. In urban areas, they found evidence that 60% of the HNCO came from primary (i.e., directly emitted) sources while 40% came from secondary (i.e., photochemically produced) sources. Wentzell et al. (2013) used a CIMS to measure ambient concentrations of HNCO in urban Toronto and found that the measured HNCO (20-140 pptv) correlated strongly with benzene, which the authors interpreted as a fossil-fuel based source. Zhao et al. (2014) measured HNCO using a CIMS at an elevated site near La Jolla,

CA and found evidence for photochemical production as well as significant uptake of HNCO by clouds. And finally, Chandra and Sinha (2016) measured ambient HNCO in Mohali, India using a proton transfer reaction mass spectrometer (PTR-MS) and found that the ambient concentrations regularly exceeded 1 ppbv during post-harvest agricultural burning. Despite only a handful of observations in a few locations, it is clear that atmospheric HNCO has both natural (e.g., wildfires) and anthropogenic (e.g., agricultural burning) sources. With these sparse observations, it is clear that we are just beginning to

understand the spatiotemporal distribution and the contribution of natural and anthropogenic sources to ambient concentrations of HNCO.

Similar to the ambient observations, there have only been a handful of studies that have investigated HNCO emissions from anthropogenic sources, most of which have focused on gasoline and diesel-powered sources. Previous HNCO studies on

natural, biomass burning sources have been performed by Roberts et al. (2011) and Coggon et al. (2016). Brady et al. (2014) measured tailpipe emissions of HNCO from eight light-duty gasoline vehicles (LDGVs) and suggested that HNCO emissions from LDGVs were not a result of in-cylinder combustion but rather a result of CO and $NO_x$ dependent chemistry in the aftertreatment device, namely the three-way catalytic converter. Wentzell et al. (2013) measured tailpipe emissions of HNCO from an on-road diesel engine but did not conclusively point to the source (in-cylinder or aftertreatment) of the

HNCO. The HNCO emissions were comparable to those from LDGVs measured by Brady et al. (2014) but varied substantially (order of magnitude or more) with the drive cycle and possibly with the co-emitted $NO_x$ emissions. Link et al. (2016) found that diesel engines could not only produce in-cylinder HNCO but also emit precursors (e.g., amides) that could photooxidize in the atmosphere to form secondary HNCO. It appears that there are large uncertainties surrounding the sources and precursors of HNCO from anthropogenic sources (e.g., in-cylinder or aftertreatment, primary or secondary) and

there is a need to perform additional studies that can help elucidate the chemistry and conditions that lead to HNCO production.

Increasingly, new diesel engines sold in developed economies (e.g., United States, Canada, European Union) need to be equipped with Selective Catalytic Reduction (SCR) systems to reduce tailpipe emissions of $NO_x$ and meet newer/stricter

emissions standards (e.g., EPA's 2010 standard for heavy-duty on-road engines, California's Drayage Truck Regulation, EPA's Tier 4 standard for non-road engines, Euro 6 standard for heavy-duty trucks and buses). In an SCR system, urea thermally decomposes to produce ammonia ($NH_3$) and HNCO ($H_2N-CO-NH_2 \rightarrow NH_3 + HNCO$ - R1) and the HNCO rapidly hydrolyzes on the catalyst surface to yield another $NH_3$ molecule ($HNCO + H_2O \rightarrow NH_3 + CO_2$ - R2); $NH_3$ is the active agent that reduces NO and $NO_2$ to $N_2$ and $H_2O$:

$2 NH_3 + NO + NO_2 \rightarrow 2 N_2 + 3 H_2O$- R3, fast SCR chemistry

$4 NH_3 + 4 NO + O_2 \rightarrow 4 N_2 + 6 H_2O$ - (R4)

$8 NH_3 + 6 NO_2 \rightarrow 7 N_2 + 12 H_2O$ - (R5)





Since SCR systems produce HNCO as an intermediate product, they have been implicated as an atmospheric source of HNCO (Roberts et al., 2011). Heeb and coworkers (Heeb et al., 2011;Heeb et al., 2012) performed experiments on an on-road diesel engine and found an order-of-magnitude increase in HNCO emissions with the SCR system engaged, implying that SCR systems could potentially be a source for HNCO. However, Heeb and coworkers (Heeb et al., 2011;Heeb et al., 5  2012) used an offline technique to measure HNCO, which lacks the time resolution and sensitivity found in online mass spectrometry instrumentation.

To date, there has only been a single study that has used a large-scale model to simulate ambient concentrations of HNCO from biomass burning and biofuel combustion. Leveraging the measurements of Roberts et al. (2011), Young et al. (2012) 10  simulated ambient concentrations of HNCO using a global model. They found that surface HNCO concentrations might only be of human health concern (>1 ppbv for more than 7 days of the year) in tropical regions dominated by biomass burning (Southeast Asia) and in developing countries (Northern India and Eastern China) dominated by biofuel combustion. Although Young et al. (2012) acknowledged that anthropogenic sources such as gasoline and diesel engines and secondary processes in the atmosphere might be important contributors to atmospheric HNCO, they did not include these 15  sources/pathways in their study. Furthermore, the grid resolution of the model used by Young et al. (2012) was too coarse (2.8° ×2.8°) to resolve elevated HNCO concentrations in urban areas. So it is not known whether anthropogenic sources other than biofuel combustion and secondary production could result in elevated levels of HNCO in urban areas affected by mobile source pollution.

20  In this work, we performed laboratory experiments to measure HNCO emissions from an SCR-equipped, modern-day, non-road diesel engine to test if SCR systems were a potential source of HNCO. To quantify HNCO emissions under different operating conditions, we performed these tests under varying urea injection rates (stoichiometric ratios of 0 to ~1.3), engine loads (idle-like, intermediate speed, rated speed) and fuels (diesel, biodiesel). The HNCO was measured using a time-of-flight, acetate-based, chemical ionization mass spectrometer. Based on findings from our and previous work, we used a 25  chemical transport model (CTM) to simulate ground-level concentrations and source (gasoline, diesel, biomass burning) and process (primary, secondary) contributions to HNCO in California.

## 2 Methods

### 2.1 Laboratory Experiments

*Engine:* The HNCO experiments were conducted on an engine dynamometer-mounted (Midwest Inductor Dynamometer 30  1014A) 4-cylinder, turbocharged and intercooled, 4.5 L, 175 hp, John Deere 4045 PowerTech Plus diesel engine; this engine platform has been part of several earlier research studies (Jathar et al., 2017;Drenth et al., 2014). The engine consisted of a variable geometry turbocharger, exhaust-gas recirculation, and electronically controlled high-pressure common rail fuel injection and met non-road Tier 3 emissions standards. A diesel oxidation catalyst (DOC, John Deere RE568883) and diesel particulate filter (DPF, John Deere RE567056) were retrofitted on the exhaust system to meet non-road interim Tier 4 35  emission standards. Recently, Jathar et al. (2017) found that the DOC+DPF retrofitted system used in this work at 50% engine load resulted in reductions of CO and particulate matter similar to those found across a compendium of on- and non-road diesel engines (May et al., 2014).

*Selective Catalytic Reduction (SCR) System:* We built and installed a custom selective catalytic reduction (SCR) system in 40  the exhaust line that allowed us to control and explore HNCO emissions as a function of varying urea injection rates; the SCR system was installed downstream of the DOC+DPF (see schematic in Figure S-1). The urea injection rates were controlled using an SCR-specific engine control unit (ECU) provided by John Deere. The SCR ECU controlled a high





pressure pump coupled to an injector to aerosolize urea into the exhaust. A baffle-based mixer downstream of the injection location facilitated homogenous mixing of the aerosolized urea with exhaust. The urea and exhaust mixture was passed over 0.492 ft³ of Cu-Zeolite catalyst, which are catalysts of choice for on-road SCR systems in the United States. The total catalyst volume for our system (0.492 ft³) was determined based on recommended values for the space velocity, which

5   quantifies the exhaust volume processed per hour.

*Engine Tests:* We performed a total of nine tests at three different engine loads (idle-like, 50% load at intermittent speed, and 50% load at rated speed) and with two different fuels (diesel and fatty acid methyl ester-based biodiesel). The three engine loads were (i) 45 Nm at 2400 RPM and 11 kW, (ii) 284 Nm at 1500 RPM and 45 kW, and (iii) 226 Nm at 2400 RPM and 57

10  kW, which corresponded to modes 4, 7, and 3 on the ISO 8178-4 C1 duty cycle respectively. The ISO 8178 duty cycle is an international standard used for emissions certification for non-road diesel engines. The diesel fuel was commercial, non-road, ultra-low-sulfur diesel (ULSD) and sourced locally while the biodiesel fuel (B100) was sourced from Blue Sun (St. Joseph, MO) and produced from soy; we have included the fuel certificate as an appendix. Each test included a sweep across three to four urea injection rates for each engine load-fuel combination. We used commonly available diesel exhaust fluid - a

32.5:67.5 mixture of urea and water, as our urea source. After changing the urea injection rate, engine load, or fuel, the emissions were allowed to stabilize for approximately ten minutes before values were recorded.

*Instruments:* Raw exhaust was transferred to a Siemens 5-gas analyzer using a 110 °C heated Teflon™ transfer line followed by a water trap to measure $CO_2$ (non-dispersive infrared), CO (non-dispersive infrared), unburned hydrocarbons (flame

ionization detector), NO and $NO_y$ (chemiluminescence) and $O_2$ (electrochemical). Raw exhaust was sampled through an isokinetic probe using 15 feet of Silcosteel® tubing heated to 150 °C and diluted with activated charcoal- and HEPA-filtered air using a Hildemann-style dilution sampler (Hildemann et al., 1989). The dilution ratios were calculated using the method outlined by Lipsky and Robinson (2006) based on $CO_2$ measurements. The diluted exhaust was diluted even further with ultra-high purity $N_2$ before being sampled by an acetate reagent ion-based, time-of-flight, chemical ionization mass

spectrometer (ToF-CIMS; Tofwerk AG and Aerodyne Research, Inc.) to measure HNCO. The operation of the CIMS and reagent-ion chemistry was similar to that described in Link et al. (2016) with minor differences. The acetic anhydride reagent source was stored in an oven and transfer lines were kept at a constant temperature of 40°C using heating tape. The ToF duty cycle was set to 16 kHz and data was acquired at 1 Hz resolution. A cross-calibration method was used to quantify HNCO similar to that described in Brady et al. (2014). The primary assumption of the cross-calibration method was that the

ratio of formic acid sensitivity to HNCO sensitivity would remain the same between the two instruments operated under similar voltage settings as described in equation 1;

$$[HNCO]_{pptv} = \frac{[CNO^-]_{(2)}F_{formic(1)}}{F_{formic(2)}F_{HNCO(1)}} - (1)$$

where *[CNO⁻]₍₂₎* is the measured *CNO⁻* ion signal normalized to the acetate reagent ion signal, $F_{formic(1)}$ is the formic acid sensitivity (pptv ncps⁻¹) measured by Link et al. (2016), $F_{formic(2)}$ is the formic acid sensitivity (pptv ncps⁻¹) measured during

this study, and $F_{HNCO(1)}$ is the HNCO sensitivity (pptv ncps⁻¹) as reported in Link et al. (2016). A formic acid calibration was performed at the beginning and end of each day of experiments to obtain $F_{formic(2)}$ using a custom built permeation oven and formic acid permeation source.

*Emission Factors:* Background-corrected emission factors (EF) for CO, NO, $NO_2$, and HNCO were calculated using

equation 1 and expressed as grams of pollutant produced per kg of fuel burned. Since more than 98% of the fuel carbon was emitted as $CO_2$, we assumed that in equation 2 all of the carbon in the fuel was converted to $CO_2$.

$$EF = \frac{[P]}{\frac{[CO_2]}{MW_{CO_2}}} \times MW_C \times C_f \times FC - (2)$$





Here, *[P]* is the background corrected pollutant concentration in µg m$^{-3}$, *[CO$_2$]* is the background corrected $CO_2$ concentration in µg m$^{-3}$, $MW_{CO2}$ is the molecular weight for $CO_2$, $MW_C$ is the atomic weight of C, and $C_f$ is the carbon mass fraction in the fuel in kg-C kg-fuel$^{-1}$. We use a $C_f$ of 0.85 for diesel and 0.77 for biodiesel (Gordon et al., 2014).

### 2.2 Chemical Transport Modeling

*Chemical Transport Model:* The UCD/CIT is a regional chemical transport model that has been extensively used to simulate the emissions, transport, chemistry, deposition and source contribution of pollutants in the lower troposphere (Hu et al., 2012;Jathar et al., 2016;Kleeman and Cass, 2001). HNCO simulations were performed for the state of California at a grid resolution of 24 km followed by a nested simulation over the South Coast Air Basin (SoCAB) domain at a grid resolution of 8 km from July 15$^{th}$ to August 2$^{nd}$, 2005. Simulations were performed for California since the state is home to the five most polluted cities in the United States for ozone and particulate matter (2016). We used the (i) CRPAQS (California Regional PM$_{10}$/PM$_{2.5}$ Air Quality Study) inventory for anthropogenic emissions, (ii) FINN (Fire Inventory for National Center for Atmospheric Research) inventory for biomass burning emissions (Wiedinmyer et al., 2011), and (iii) MEGAN (Model of Emissions of Gases and Aerosols from Nature) model for biogenic emissions (Guenther et al., 2006). Hourly meteorological fields were produced using the Weather Research and Forecasting (WRF) v3.4 model (www.wrf-model.org). Initial and hourly varying boundary conditions were based on the results from the global model MOZART-4/NCEP (Emmons et al., 2010). Gas-phase chemistry was modeled using SAPRC-11. For more details, we refer the reader to previous model applications (Jathar et al., 2015;Jathar et al., 2016).

*Primary Emissions of HNCO:* Primary emissions of HNCO were calculated by first determining a source-specific HNCO:CO ratio (see Table 1) and then combining them with source-specific, spatiotemporally resolved CO emissions to build an inventory for HNCO emissions. We use a ratio based approach rather than an emission factor based approach because only a handful of sources have been characterized for HNCO emissions in previous studies and, in our view, the data may not be representative enough to develop an HNCO inventory using emission factors and fuel activity data. For the same reason, we assume equivalence between on- and off-road engine sources in developing a source-specific HNCO:CO ratio. We considered three sources for primary emissions of HNCO: (1) on- and non-road diesel, (2) on- and non-road gasoline and (3) biomass burning (includes residential wood combustion). HNCO:CO ratios for diesel sources were determined based on the range of measured HNCO:CO ratios found in this and previous work. Findings from this work (see Sections 4.1 and 4.2) suggest that none of the aftertreatment systems deployed on our diesel engine affected HNCO emissions but the DOC dramatically reduced CO emissions (factor of ~30 at 50% load for the engine described herein). In other words, for the same engine we anticipate that the presence of a DOC will increase HNCO:CO ratios by a factor of 30 assuming that the HNCO emissions do not change with the DOC. Hence, we need to be careful in how we calculate the HNCO:CO ratio and also how the HNCO:CO ratio is applied to determine primary emissions of HNCO. Since we are modeling an episode prior to when diesel engines were required to have a DOC, we have only used non-DOC data to calculate low and high estimates for HNCO:CO ratios. Based on this work, and that of Link et al. (2016), Heeb and coworkers (Heeb et al., 2011;Heeb et al., 2012) and Wentzell et al. (2013), we loosely calculated a lower bound HNCO:CO ratio of ~0.001 that reflected diesel engine operation at lower engine loads and a higher HNCO:CO ratio of ~0.01 that reflected diesel engine operation at higher engine loads; the HNCO:CO data from all sources are tabulated in Table S-1. The HNCO:CO ratio for the gasoline sources was determined as the ratio of the median HNCO to the median CO measured by Brady et al. (2014) for eight light-duty gasoline vehicles. The HNCO:CO ratio for biomass burning sources, which includes residential wood combustion, was based on an approximate fit to the laboratory and ambient data measured by Veres et al. (2010), Roberts et al. (2011) and Yokelson et al. (2013). The HNCO:CO ratios were then combined with source-specific, spatiotemporally resolved CO emissions to build source-resolved emissions for HNCO. HNCO from the three sources were





tracked separately in the UCD/CIT model. We note that the previous study that simulated HNCO in a 3D model developed global emissions of HNCO by using a source-specific ratio of HNCO with hydrogen cyanide (Young et al., 2012).

*Photochemical Production of HNCO:* Link et al. (2016) observed strong photochemical production of HNCO from a diesel
engine without any aftertreatment. This secondary HNCO source can be attributed to photooxidation of amides (e.g., formamide, acetamide) and potentially other reduced organic nitrogen compounds present in the diesel exhaust, though the full suite of precursors, their reaction mechanisms and their HNCO yields remains unknown. Hence, we make simplifying assumptions to parameterize photochemical production of HNCO in our CTM simulations. We assumed that diesel exhaust contains a single HNCO precursor ($X$) that reacts with the hydroxyl radical ($OH$) to form HNCO, i.e., $X + OH \rightarrow HNCO$.
Assuming that the emissions for $X$ scale with primary emissions of HNCO, and $X$ and $OH$ participate in a first order reaction (i.e., $X = X_0 e^{-k_{OH}[OH]\Delta t}$), the emissions for $X$ and its reaction rate with OH ($k_{OH}$) can be determined from a fit to the experimental data from Link et al. (2016). Separate parameterizations for emissions of $X$ were developed for the two engine loads (idle and 50% load at rated speed) described in Link et al. (2016). Fits and the parameters are shown in Figure 1. The diesel and biodiesel data were nearly identical and hence data from both fuels were used to determine the engine load-
specific fits. The physical interpretation of the fit for idle conditions is that for 1 kg of fuel burned, ~0.050±0.006 g of HNCO and ~0.20±0.01 g of $X$ are emitted, with $X$ reacting with $OH$ with a $k_{OH}$ of 5.5±1.3×10$^{-12}$ cm$^3$ molecules$^{-1}$ s$^{-1}$ to form HNCO. The fit values for the precursor scaling with respect to the primary emissions of HNCO, i.e., 4.9±0.3 at idle conditions and 2.2±0.2 at 50% load conditions are 2.9 and 6.4 times lower and the fit $k_{OH}$ value is approximately two times higher than that calculated by Roberts et al. (2014) from ambient observations (precursor scaling of 14.1 and $k_{OH}$ of 2.33×10$^{-}$
$^{12}$ cm$^3$ molecules$^{-1}$ s$^{-1}$). We note that our precursor scaling and $k_{OH}$ fits are not unique and that we could produce a higher (or lower) precursor scaling and a corresponding lower (or higher) $k_{OH}$ pair that would fit the data equally well and possibly align better with the fits from Roberts et al. (2014). Our fit value of $k_{OH}$ compares quite well with that calculated by Borduas et al. (2014) for formamide (4.44±0.46×10$^{-12}$ cm$^3$ molecules$^{-1}$ s$^{-1}$); Borduas et al. (2014) observed HNCO production from OH oxidation of formamide. Spatiotemporally resolved precursor emissions were developed for diesel sources by
multiplying the primary HNCO emissions developed in the previous section by the scaling (i.e., 4.9 and 2.2) determined through the fits. The reaction chemistry to form secondary HNCO was added to SAPRC-11. We do not consider HNCO precursors for gasoline and biomass burning sources.

*Loss Mechanisms for HNCO:* Roberts et al. (2011) argued that the atmospheric reaction of HNCO with OH and photolysis
of HNCO were too slow to be relevant in the atmosphere, and claimed that the only relevant loss mechanism for HNCO was dry and wet deposition. Young et al. (2012) investigated the influence of heterogeneous uptake of HNCO by clouds in a global model and found that this loss mechanism competed with dry deposition only when the cloud *pH* was 6 or higher. In a follow up study, Barth et al. (2013) used a detailed box model to suggest that HNCO concentrations could be significantly depleted if air parcels containing HNCO encountered low-level cumulus clouds. To simplify the treatment in the CTM and
provide an upper bound on our HNCO estimates, we only modeled dry deposition for HNCO assuming that HNCO was equivalent to nitric acid. We chose nitric acid since, like HNCO, nitric acid is extremely soluble in water at neutral *pH*. The HNCO precursor was modeled as NO to determine its dry deposition.

### 3 Results and Discussion

### 3.1 NOₓ and HNCO Emission Factors

In Figure 2, we plot emission factors for NO$_x$ and HNCO as a function of the NH$_3$ injection rates for all the experiments performed during our study. NH$_3$ injection rates were calculated by assuming that all of the urea thermally decomposed into



$NH_3$. As expected, we saw a near-exponential decrease in $NO_x$ emissions with a linear increase in $NH_3$ injection. Within the calculated range of stoichiometric doses for $NH_3$, $NO_x$ emissions for most engine load-fuel combinations tested in this study were reduced by more than 90%; stoichiometric doses of $NH_3$ were calculated assuming all of the $NO_x$ was either NO or $NO_2$ and followed reactions R4 and R5 respectively. The only exceptions were the idle-like load experiments (2400 RPM

and 11 kW) where we could not inject more $NH_3$ (>~0.012 g s$^{-1}$) since higher injections lowered the catalyst temperatures to values below those required for normal functioning of the SCR (<200 °C). We also found that the $NO_x$ emissions continued to decrease beyond stoichiometric injections of $NH_3$. This allowed the SCR-equipped diesel engine to meet and exceed the most recent EPA Tier 4 emission standard of ~1.6 g of $NO_x$ kg-fuel$^{-1}$ (or 0.4 g of $NO_x$ kW-hr$^{-1}$). Similar to earlier work (Wentzell et al., 2013;Heeb et al., 2012;Heeb et al., 2011), we observed HNCO when no $NH_3$ was injected, implying that the

HNCO was produced either in the engine cylinder or in the aftertreatment devices upstream of the SCR (DOC+DPF). For six of the seven experiments performed with diesel fuel, HNCO emissions were reduced by 5-40% with increasing $NH_3$ injections. For one of the diesel experiments and both the biodiesel experiments in which the measured emission factors (and measured concentrations) for HNCO were the lowest, HNCO emissions increased by 30-125% as the $NH_3$ injection was increased. In summary, it is unlikely that SCR-equipped engines running on diesel fuel are a source of HNCO even when the

$NH_3$ injection exceeds stoichiometric rates. It is possible that the use of biodiesel reduces primary emissions of HNCO and that the HNCO enhancements in these experiments reflect slight contributions from the SCR chemistry that are undetectable during most of the diesel experiments. The HNCO response on SCR systems when using biodiesel needs to be explored in future work.

### 3.2 Intercomparison with Earlier Work

Very few studies have investigated HNCO emissions from diesel engines and, before this work, only one study had examined HNCO emissions from an SCR-equipped diesel engine. In Figure 3, we compare HNCO emission factors for diesel fuel with all earlier work involving diesel engines: Link et al. (2016), Wentzell et al. (2013), Heeb et al. (2011) and Heeb et al. (2012). We also compare our results to HNCO emissions from an ensemble average of eight light-duty gasoline vehicles measured by Brady et al. (2014). For this work, the mean and the standard errors were calculated using the HNCO

data across all $NH_3$ injection rates. The HNCO emission factors across these five studies spanned nearly three orders of magnitude and while providing some insight, highlight the uncertainty in both the emissions and the measurements of HNCO. First, the emission factors for HNCO from this work were nearly identical to those measured by Link et al. (2016). Since both studies were performed on the same engine and used a similar CIMS instrument, the HNCO was mostly likely produced in the engine cylinder and was unaltered by the DOC and DPF. Second, primary emissions of HNCO and its

precursors based on the work of Link et al. (2016) were deemed plausible when compared against emissions of total unburned hydrocarbons, i.e., HNCO and its precursor at idle conditions were less than 0.2% and 1% of the total hydrocarbon emissions while HNCO and its precursor at 50% load conditions were less than 0.4% and 0.9% of the total hydrocarbon emissions. Third, the emission factors for HNCO from the engine used in this work (with or without the aftertreatment devices) were much higher (factor of 10-100) than those measured by Wentzell et al. (2013). Wentzell et al. (2013) used a

CIMS instrument similar to that used in this study and therefore the differences could not be attributed to the instrumentation. Link et al. (2016) suggested that the large differences between their study and the Wentzell et al. (2013) study could reflect the variability found in emissions between non- and on-road diesel engines and steady and transient drive cycles. However, when compared using the HNCO:CO ratio, there was much less variability in the ratio between this work and two of the drive cycles examined by Wentzell et al. (2013) (see Table S-1), which could suggest that our non-road

engine, on account of being larger than the Wentzell et al. (2013) engine, simply produced more HNCO and more CO but yielded the same HNCO:CO ratio. This observation led us to assume (in Section 3.2) equivalence between non-road and on-road diesel engines as well as to develop emissions inventories for HNCO based on the HNCO:CO ratio, rather than through




the use of emission factors. To test our findings and assumptions, we recommend that future studies focus on testing a diverse suite of diesel engine sizes under a wide range of steady and transient engine loads. Fourth, the emission factors for HNCO from this study (31-56 mg kg-fuel$^{-1}$) compared reasonably well with those from an SCR-equipped diesel engine tested by Heeb and coworkers (Heeb et al., 2011;Heeb et al., 2012) (29-32 mg kg-fuel$^{-1}$). However, in sharp contrast to our

findings from Figure 2, Heeb and coworkers (Heeb et al., 2011;Heeb et al., 2012) saw dramatically reduced (factor of 10) HNCO emissions without the SCR and suggested that the SCR was a source of HNCO. Heeb and coworkers (Heeb et al., 2011;Heeb et al., 2012) employed an offline technique to measure HNCO (liquid-phase sample collection followed by hydrolysis of HNCO to $NH_3$ and measurement of $NH_3$) and it is possible that differences in the HNCO emission factors reflect a change in the sensitivity of the technique to changes in $NH_3$ concentrations in the tailpipe during SCR system

operation. And finally, the emissions of HNCO on a fuel-burned basis from this work and Roberts et al. (2011) were more than an order of magnitude larger than the average HNCO emissions from the suite of light-duty gasoline vehicles tested by Brady et al. (2014). Thus, despite higher gasoline consumption in the United States compared to diesel (~3:1), diesel engines, regardless of their use of aftertreatment devices, might be a much larger source of HNCO than catalytic-converter-equipped gasoline engines.

### 3.3 Chemical Transport Model Results

Predictions of 14-day averaged, ground-level concentrations of HNCO from the CTM simulations are mapped for the state of California in Figure 4(a-b). The low and high results are from two simulations that used two different primary emissions and photochemical production parameterizations for diesel engines (refer to Table 1 for details). Inland concentrations of HNCO between the low and high simulations varied significantly but never exceeded 110 pptv and were at least an order of

magnitude lower than the 1 ppbv level proposed by Roberts et al. (2011). The highest concentrations of HNCO from the high simulation were found in Los Angeles (low estimate=20.1 pptv, high estimate=107 pptv) located in the South Coast Air Basin (SoCAB); SoCAB is home to 17 million people and consistently the most polluted in the United States for ozone and particulate matter. HNCO concentrations from the high simulation in four other locations (Riverside, Fresno, Bakersfield, and Sacramento), where ozone and particulate matter concentrations are amongst the worst in the country, varied between 23

and 66 pptv. We individually tracked the source/process-level contributions of HNCO in the CTM simulations and found that diesel use was the dominant source of HNCO in SoCAB. Based on the low and high simulations, diesel sources accounted for 55-92% while gasoline sources accounted for 8-41% of the HNCO in SoCAB, with a very small contribution (1-4%) from biomass burning sources. The signature of a larger contribution of HNCO from biomass burning sources can be seen in Figure 4(a) in more remote locations of California, e.g., north-west corner of California, north of Sacramento.

Despite the strong photochemical production observed by Link et al. (2016) in laboratory experiments, secondary production of HNCO from precursors in diesel exhaust only accounted for 9-11% of the total HNCO. The most likely explanation for this small contribution was that the in-basin exposure of HNCO precursors to OH radicals was too small to produce a lot of secondary HNCO. In fact, the slow secondary production of HNCO can be visualized in Figure 1 where significant enhancements in HNCO were only observed after ~5 hours of photochemical processing. We investigated the sensitivity of

model predictions to dry deposition by using NO as the surrogate to model dry deposition of HNCO; NO has a much slower dry deposition lifetime than nitric acid and we expected HNCO concentrations to increase. Model predictions suggested that HNCO concentrations increased by ~50% in urban areas and by a factor of 2 to 5 in rural/remote regions.

### 3.4 Model Evaluation

To evaluate the HNCO predictions from our CTM simulations, we compared model predictions from the 8 km simulation to

two datasets of HNCO measurements in urban areas: (i) observations reported by Roberts et al. (2014) at the Pasadena ground site during the California Research at the Nexus of Air Quality and Climate Change (CalNex) study in May-June





2010 and (ii) observations reported by Wentzell et al. (2013) in Toronto in September-October 2012. Since the model simulations were not for the same time period (in the case of CalNex) or the same location (in the case of Toronto), we present the comparisons in Figure 4(c) by regressing concentrations of HNCO against those of benzene. Model predictions from the low simulation seemed to agree with the nighttime observations of Roberts et al. (2014) and validate the primary

parameterizations and deposition scheme used in the low simulation. Roberts et al. (2014) have argued that the diurnal differences in the observations imply a daytime photochemical source of HNCO, where in the mid-afternoon secondary processing accounts for 40% of the total HNCO. Hence, agreement between the model predictions from the high simulation and the daytime observations of Roberts et al. (2014) should not be construed as a validation of the inputs for that simulation since the high simulation predicts a small contribution (~10%) of HNCO from photochemical production. One interpretation

of this model-measurement comparison is that the model is missing HNCO precursors from gasoline and biomass burning sources that may lead to increases in daytime production of HNCO. In contrast to the comparison at Pasadena, the model predictions from the high simulation compared well with rush hour observations of primary HNCO by Wentzell et al. (2013), which could be seen as a validation of the primary parameterizations in the high simulation. As is clear, the model evaluation is severely limited because of lack of laboratory data sets that can help parameterize the emissions and chemistry

of HNCO and lack of observational data sets that can help validate those parameterizations. Nonetheless, the range of HNCO concentrations predicted between the low and high simulations are bound by the two observational datasets and hence, this work provides a reasonable set of parameterizations to model HNCO in CTMs.

## 4. Summary and Conclusions

We performed laboratory experiments on an SCR-equipped modern day diesel engine to measure emissions of isocyanic

acid (HNCO) as a function of varying urea injection rates, engine loads, and fuels. We found no evidence that the SCR or the other aftertreatment devices (diesel oxidation catalyst and diesel particle filter) were a source of tailpipe HNCO. We argue that the HNCO from diesel engines was likely produced inside the engine cylinder during fuel combustion. This finding is not completely new. Chemical kinetics models (Mansour et al., 2001), model systems with propane (Nelson and Haynes, 1994), and engine tests without aftertreatment devices (Heeb et al., 2011) have previously shown that HNCO (and other

reduced nitrogen-containing compounds) can be produced during combustion in the presence of $NO_x$. We note that the exhaust gas recirculation (EGR) system that adds $NO_x$-containing exhaust to the engine to reduce cylinder temperature and consequently reduce $NO_x$ production may actually enhance in-cylinder HNCO production from the increased homogenous availability of $NO_x$; the engine employed in this work and most of those found in the United States on mobile sources have EGRs. EGRs as an enabler of in-cylinder HNCO production needs to be explored in future studies.

Amides such as formamide are known precursors of HNCO (Borduas et al., 2014) and might be part of amide emissions from various types of combustion sources that lead to atmospheric production of HNCO. Recent studies have noted that other forms of reduced organic nitrogen compounds can be oxidized to form HNCO, suggesting that molecules other than amides emitted from diesel exhaust may also be HNCO precursors (Borduas et al., 2016a;Borduas et al., 2016b). Fits to the

data from Link et al. (2016) suggest that the emissions and the rate of photochemical production of HNCO attributed to diesel sources may not be sufficient to contribute significantly to ambient concentrations of HNCO in urban environments. These precursors, however, might be important in controlling HNCO concentrations in remote/rural environments but based on the results from this study might be deemed too low to be of any concern from a health perspective. Our chemical transport model (CTM) predictions suggest that the daily-averaged precursor concentrations in urban environments are large

enough (70-300 pptv in Los Angeles, see Figure S-2 for precursor concentrations in California) to provide impetus for ambient studies to design and deploy instruments to measure these precursors. Finally, it is possible that sources other than




diesel engines (gasoline engines, biomass burning) also emit precursors of HNCO and hence need to be studied in the future both in terms of identifying and quantifying the precursors of HNCO and measuring their potential to form HNCO.

We used a chemical transport model (CTM) to simulate ground-level concentrations and source (gasoline, diesel, biomass burning) and process (primary, secondary) contributions to HNCO in California. The predicted HNCO concentrations in Southern California were roughly similar to those measured at Pasadena in 2010, Toronto in 2012, and La Jolla in 2012. A detailed comparison at Pasadena highlighted missing precursors/pathways for photochemical production of HNCO during the daytime. The comparisons also implied that diesel engines (and possibly gasoline engines) are large sources of HNCO in urban areas. In the simulations, daily-averaged HNCO concentrations never exceeded 110 pptv and were an order of magnitude below the 1 ppbv level that Roberts et al. (2014) have proposed could result in human health effects. If we assume that the HNCO-benzene regression from our work holds for other parts of the world, benzene concentrations exceeding 7 ppbv would be associated with 1 ppbv levels of HNCO. We should note that the 1 ppbv threshold is a rough estimate and we see a need for epidemiological and/or toxicological studies that would better inform that estimate. Emissions from biomass burning sources in the winter combined with a strong likelihood for temperature inversions could lead to higher HNCO concentrations in the winter and need to be explored using both measurements and air quality modeling.

**Acknowledgements**

We would like to thank Daniel Olsen for inputs on experimental design, Kirk Evans for technical support and undergraduate researcher Liam Lewane for engine test support during the study. We would also like to thank DCL International Inc. for donating SCR catalysts for our study. D. K. Farmer acknowledges an Arnold and Mabel Beckman Young Investigator Award for funding the laboratory HNCO measurements.

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



**Tables**

Table 1: Emissions and emissions ratios for gasoline, diesel and biomass burning sources for the state of California for an average summer day in 2005.

| Source | CO emissions (tons day$^{-1}$) | HNCO:CO | HNCO (tons day$^{-1}$) | HNCO Precursor (tons day$^{-1}$) |
|---|---|---|---|---|
| Gasoline | 8195 | 0.000036[a] | 0.29 | 0 |
| Diesel (low) based on idle data | 447 | 0.001[b] | 0.45 | 2.21 |
| Diesel (high) based on load data | 447 | 0.01[b] | 4.47 | 9.83 |
| Biomass Burning | 164 | 0.001[c] | 0.16 | 0 |

[a]Brady et al. (2014); [b]This work, Link et al. (2016), Heeb et al. (2011), and (Heeb et al., 2012); [c]Roberts et al. (2011)

5 **Figures**

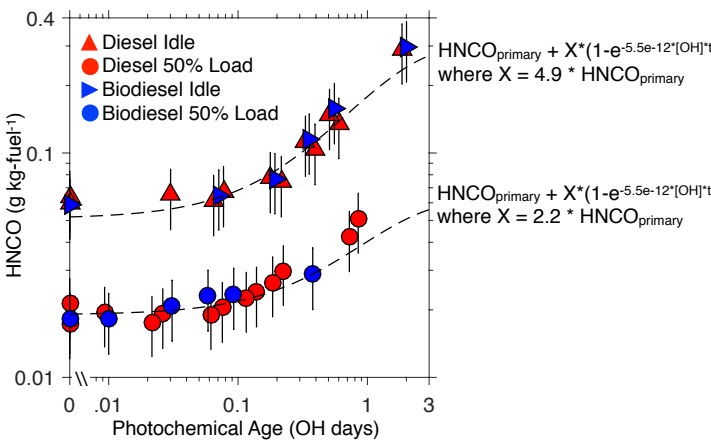

Figure 1: Emission/production factors for HNCO as a function of photochemical age from Link et al. (2016). The fitting, which parameterize the emissions of the HNCO precursor and the reaction rate constant with OH, has been performed in this work. The photochemical age is calculated assuming an OH concentration of $1.5 \times 10^6$ molecules cm$^{-3}$.

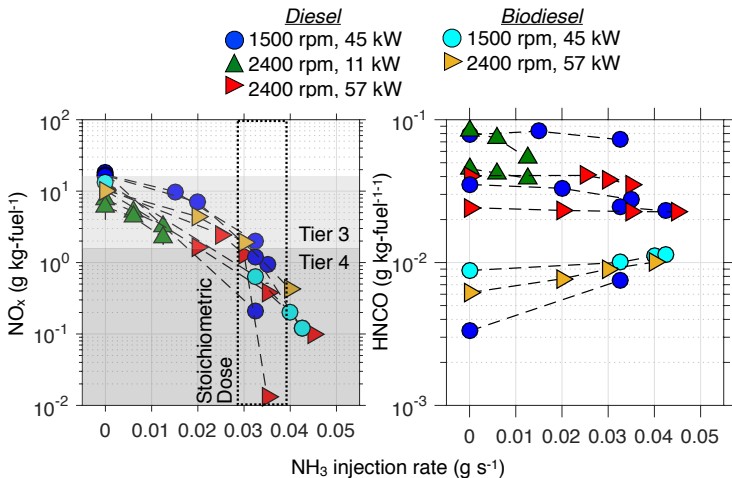

Figure 2: Emission factors for (a) NO$_x$ (NO+NO$_2$) and (b) HNCO with varying NH$_3$ injection rates for all the experiments performed in this work. The NH$_3$ injection rates are calculated assuming each urea molecule produces two NH$_3$ molecules.


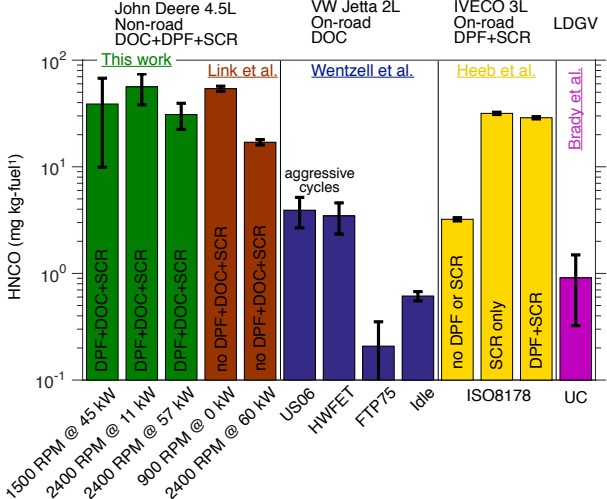

Figure 3: Emission factors for HNCO from this work compared to literature data. US06, HWFET, FTP75, ISO8178 and UC are vehicle/engine drive cycles.

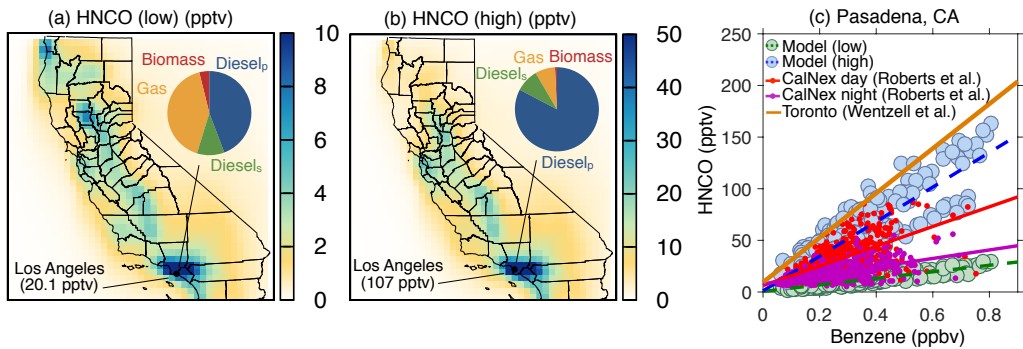

Figure 4: (a) Low and (b) high estimates of the 14-day averaged ground-level concentrations of HNCO from the 24 km simulations. (c) 14-day averaged concentrations of HNCO regressed against benzene over the South Coast Air Basin from the 8 km simulations; also includes observations from Wentzell et al. (2013) for Toronto-2012 and Roberts et al. (2014) and Borbon et al. (2013) from Pasadena-2010.