# Peer review of "Investigating Diesel Engines as an Atmospheric Source of Isocyanic Acid in Urban Areas"

_Atmospheric Chemistry and Physics, 2017_

## Referee Comment (RC1) · Anonymous Referee #1 · 24 Mar 2017

General Comments:

Jathar et al present new measurements of isocyanic acid (HNCO) emissions from diesel engines to assess the role of selective catalytic reduction (SCR) systems in enhancement of HNCO emissions. The authors demonstrate, as one would expect, that NOx emissions are reduced when SCR systems are in place, but surprisingly SCR appears to have little impact on HNCO emissions. The authors take emission ratios (HNCO:CO) determined for diesel emissions, alongside emission ratios from other primary sources and estimates of secondary production to assess HNCO concentrations in a regional model. Both elements of this study of novel contributions to the literature. The paper is well written and should be published in ACP following the author's attention to the following comments:

Specific Comments:

1) There is a paper by Suarez-Bertoa and Astorga, Isocyanic acid and ammonia in vehicle emissions, in Transportation Research Part D: Transport and the Environment, which is not cited which has some discussion of HNCO emissions and SCR that are likely relevant to this discussion.

2) It would be helpful for the authors to provide more discussion on the choice and potential implications of using HNCO:CO vs EFHNCO. As I understand this is done because an emission factor for biomass burning is not trivial. However, I would like to know more about the implications of this decision for the gasoline and diesel emissions. If one were to implement an emissions factor based approach would you expect the same conclusions (e.g., that diesel emissions for HNOC are more important than gasoline emissions and that [HNCO] are routinely less than 100 ppt in urban areas?

3) The paper concludes that SCR does not enhance HNCO emission factors. However, I am struggling to see this so clearly in Figure 2. There is a tremendous amount of variability in each of the data sets. For example the blue dots (1500 rpm) span almost 2 orders of magnitude when [NH3] is zero? The reduction in NOx (Fig 2A) is very clear, but my interpretation of Fig. 2B would be that HNCO emissions when SCR is used do not change within an order of magnitude. Within the confidence limits of the data set, can it really be concluded that SCR does not impact HNCO emission factors? Perhaps I am missing something.

4) I was intrigued that the modeled dry deposition velocity of HNCO was taken to be equal to HNO3. I think it would be helpful to state what the corresponding HNCO lifetime is in the model wrt/deposition and how much this assumption impacts model [HNCO]. It is easy to imagine a factor of 2 if not much more uncertainty in this assumption. It would be helpful to the reader to know how important this term is.

5) I understand that benzene and HNCO should be strongly correlated near the source region, but these two molecules have very different atmospheric lifetimes. It would be

helpful for the authors to provide some comment on the limits of making such correlations for non-source regions.

---

## Referee Comment (RC2) · Anonymous Referee #2 · 27 Mar 2017

This manuscript concerns the potential for diesel engines to be a source of isocyanic acid (HNCO), particularly when they are used with urea selective catalytic reduction (SCR) systems. The study is of relevance as HNCO has been mooted as a potential driver of negative health outcomes.

The work is in two parts: 1) laboratory experiments to determine the emissions of HNCO from a test engine fitted with an SCR system, with different treatments of urea (the reducing agent), and running the engine at different loads; and 2) regional chemical transport model (CTM) simulations to probe the expected ambient HNCO concentrations arising from transport sources, given the emission factors determined in part 1 (over California).

For part 1, despite the potential for urea-SCR systems to produce HNCO, the authors

report no enhancement of HNCO in the tailpipe emissions. The authors do find, however, a base level of HNCO emission from their system, as well as an emission of an HNCO photochemical precursor (likely an amide), whose emission they are able to estimate. From the emission factors they determine, they estimate that ambient HNCO concentrations over California maximize at around 20-100 pptv (in the LA Basin), depending on assumptions related to deposition of HNCO.

Overall, I think that this study is well done, well written, and a valuable addition to the growing HNCO literature, providing an impetus to others to investigate the sources, fate and impacts of this compound further. I would be happy to recommend publication of this study after consideration of my minor points below.

Minor points/Technical corrections

General: There are lots of long paragraphs throughout. Please consider breaking them up to make reading the study less daunting!

P2, L14: Stylistic note - have used "it is clear" in two consecutive sentences

P2, L37 (and throughout): Please take care that the reactions/reagents are formatted well (i.e. not italics)

P3, L14: There are other limitations of the Young et al. study, such as assuming het chem is an irreversible sink. Consider bringing in the Barth et al. (2013) study at this point.

P6, L1: Any comment on what your emissions might be if you scaled them with HCN? (Also, I believe Young et al. effectively scaled theirs with CO for non biomass burning emissions since the HCN emissions were scaled with CO)

P7, L19: "Comparison with earlier work" (no need for "inter"?)

P7, L23: Is "ensemble" needed?

P8, L15-: For the model sections... I appreciate that this is a basic model study, but

it would be worth saying a bit more about the other uncertainties when modeling. E.g., uncertainties in emissions/chemistry of species important for OH, uncertainties in transport schemes, lack of (?) het chemistry, etc.

P8, L20: Some brief reminder of the context of the 1 ppbv value would be good here

P9, L3: Please explain the use of the benzene correlation

P10, L1: Is agriculture also a source of HNCO precursors?

P10, L4: "Using our experimentally-determined emission factors, we used a chemical..."

P13, Fig 1 caption: "The fits, which parameterize..."

P14, Fig 4 caption: Explain "low" and "high", say that Roberts et al. (2014) refers to CalNex (as per the panel), and please fix the color bars - how can LA be on the scale if the color bars stop at ∼50% of the color bar value? (Should there be a triangle to indicate above 10 or 100? Or perhaps a note to say that the color bar saturates)

―――――――――――――――――――――

---

## Author Comment (AC1) · 18 May 2017

We thank both reviewers for their comments. We have revised the manuscript based on their comments and queries and provided a point-by-point response below. Reviewer comments are in regular black, our response is in blue, and the additions/updated text from the manuscript are in red.

**Reviewer 1**

Jathar et al present new measurements of isocyanic acid (HNCO) emissions from diesel engines to assess the role of selective catalytic reduction (SCR) systems in enhancement of HNCO emissions. The authors demonstrate, as one would expect, that NOx emissions are reduced when SCR systems are in place, but surprisingly SCR appears to have little impact on HNCO emissions. The authors take emission ratios (HNCO:CO) determined for diesel emissions, alongside emission ratios from other primary sources and estimates of secondary production to assess HNCO concentrations in a regional model. Both elements of this study of novel contributions to the literature. The paper is well written and should be published in ACP following the author's attention to the following comments:

[Figure]

1. There is a paper by Suarez-Bertoa and Astorga, Isocyanic acid and ammonia in vehicle emissions, in Transportation Research Part D: Transport and the Environment, which is not cited which has some discussion of HNCO emissions and SCR that are likely relevant to this discussion.
We thank the reviewer for pointing this out. We have included a discussion of the Suarez-Bertoa and Astorga (2016) findings in the introduction ("Suarez-Bertoa and Astorga (2016) have measured tailpipe emissions of HNCO from a suite of modern on-road gasoline and diesel vehicles and found that the gasoline vehicles produced more HNCO than the diesel vehicles.") and the results section ("Fourth, the emission factors for HNCO from this study (31-56 mg kg-fuel-1) compared reasonably well with those from an SCR-equipped diesel engine tested by

Heeb and coworkers (Heeb et al., 2011;Heeb et al., 2012) (29-32 mg kg-fuel-1) but were slightly higher than the two SCR-equipped diesel vehicles tested by Suarez-Bertoa and Astorga (2016) (1.3-9.7 mg kg-fuel-1). However, in sharp contrast to our findings from Figure 2, Heeb and coworkers (Heeb et al., 2011;Heeb et al., 2012) saw dramatically reduced (factor of 10) HNCO emissions without the SCR and suggested that the SCR was a source of HNCO. Suarez-Bertoa and Astorga (2016), on the other hand, found that the HNCO emission factors were higher for the two SCR-equipped vehicles than the non-SCR vehicle equipped with a DOC… And finally, the emissions of HNCO on a fuel-burned basis from this work and Link et al. (2016) were more than an order of magnitude larger than the average HNCO emissions from the suite of light-duty gasoline vehicles tested by Brady et al. (2014) but similar to those measured by Suarez-Bertoa and Astorga (2016) for a range of light-duty gasoline vehicles. Suarez-Bertoa and Astorga (2016) have argued that their emission factors for HNCO were higher than those measured by Brady et al. (2014) because of differences in sampling tailpipe versus diluted emissions. Assuming the Brady et al. (2014) data are more atmospherically-relevant, diesel engines, regardless of their use of aftertreatment devices, might be a much larger source of HNCO than catalytic-converter-equipped gasoline engines despite higher gasoline consumption in the United States compared to diesel (~3:1).") and also included their results for gasoline and diesel vehicles in Figure 3 (see above).

2. It would be helpful for the authors to provide more discussion on the choice and potential implications of using HNCO:CO vs EFHNCO. As I understand this is done because an emission factor for biomass burning is not trivial. However, I would like to know more about the implications of this decision for the gasoline and diesel emissions. If one were to implement an emissions factor based approach would you expect the same conclusions (e.g., that diesel emissions for HNCO are more important than gasoline emissions and that [HNCO] are routinely less than 100 ppt in urban areas?

We did consider the choice between using HNCO:CO ratios and using an emission factor of HNCO to build an emissions inventory for HNCO. We have explained our choice by adding the following text: "Primary emissions of HNCO were calculated by first determining a source-specific HNCO:CO ratio (see Table 1) and then combining them with source-specific, spatiotemporally resolved CO emissions to build an inventory for HNCO emissions. We used a ratio based approach rather than an emission factor based approach for the following reasons. First, to our knowledge, there were no available HNCO emission factors for biomass burning. Second, there was large variability in the measured HNCO emission factors for both gasoline and diesel engines across different studies (this is illustrated in Figure 2 and discussed in Section 3.2), which presumably arose from differences in engine sizes and technology. And finally, only a handful of sources have been characterized for HNCO emissions in previous studies and, in our view, the data may not be representative enough to develop an HNCO inventory using emission factors and fuel activity data. For the same reason, we assumed equivalence between on- and non-road engine sources in developing a source-specific HNCO:CO ratio.".

We are cognizant of the uncertainty inherent in the estimation of HNCO emissions, particularly for diesel sources. To capture this uncertainty, we developed a low and a high emissions estimate for diesel sources. Our justification for the low and high estimates was provided in the paragraph on *Primary Emissions of HNCO:* "HNCO:CO ratios for diesel sources were determined based on the range of measured HNCO:CO ratios found in this and previous work. Findings from this work (see Sections 4.1 and 4.2) suggest that none of the aftertreatment systems deployed on our diesel engine affected HNCO emissions but the DOC dramatically reduced CO emissions (factor of ~30 at 50% load for the engine described herein). In other words, for the same engine we anticipate that the presence of a DOC will increase HNCO:CO

ratios by a factor of 30 assuming that the HNCO emissions do not change with the DOC. Hence, we need to be careful in how we calculate the HNCO:CO ratio and also how the HNCO:CO ratio is applied to determine primary emissions of HNCO. Since we are modeling an episode prior to when diesel engines were required to have a DOC, we have only used non-DOC data to calculate low and high estimates for HNCO:CO ratios. Based on this work, and that of Link et al. (2016), Heeb and coworkers (Heeb et al., 2011;Heeb et al., 2012) and Wentzell et al. (2013), we loosely calculated a lower bound HNCO:CO ratio of ~0.001 that reflected diesel engine operation at lower engine loads and a higher HNCO:CO ratio of ~0.01 that reflected diesel engine operation at higher engine loads; the HNCO:CO data from all sources are tabulated in Table S-1.".

Both estimates (see Figure 4) suggest that diesel engines are probably the most likely source of HNCO in urban environments. Hence, within the scope of this work (investigating diesel vehicles as an atmospheric source of HNCO), we would expect to make the same conclusions about the relative importance between gasoline and diesel vehicles as a source of HNCO.

3) The paper concludes that SCR does not enhance HNCO emission factors. However, I am struggling to see this so clearly in Figure 2. There is a tremendous amount of variability in each of the data sets. For example the blue dots (1500 rpm) span almost 2 orders of magnitude when [NH3] is zero? The reduction in NOx (Fig 2A) is very clear, but my interpretation of Fig. 2B would be that HNCO emissions when SCR is used do not change within an order of magnitude. Within the confidence limits of the data set, can it really be concluded that SCR does not impact HNCO emission factors? Perhaps I am missing something.

We agree with the reviewer that there is a large amount of variability in the HNCO emission factors. However, in Figure 2, if we were to discount the 1500 rpm-45 kW experiment that produced very little HNCO emissions (we were unable to find any reason why this experiment could have been an outlier), the data appear to group together and the uncertainty in the data (despite the small sample size) is only about a factor of two (and not an order of magnitude). The diesel data show that in all experiments except one (one of the 1500 rpm-45 kW experiments) HNCO emissions go down with SCR operation. In contrast, the biodiesel data show lower primary emissions of HNCO but increased HNCO emissions with SCR operation. Based on those observations, we have concluded that (a) diesel engines produce HNCO, possibly from in-cylinder combustion, (b) SCR systems are unlikely to be a source of HNCO, and (c) biodiesel use may decrease in-cylinder HNCO emissions. In Section 3.1, we have conceded that the increase in HNCO emissions for the biodiesel experiments might reflect slight contributions from the SCR chemistry that remained undetectable during most of the diesel experiments. All of these points have been captured in the paragraph in Section 3.1: "Similar to earlier work (Wentzell et al., 2013;Heeb et al., 2012;Heeb et al., 2011;Suarez-Bertoa and Astorga, 2016), we observed HNCO when no NH3 was injected, implying that the HNCO was produced either in the engine cylinder or in the aftertreatment devices upstream of the SCR (DOC+DPF). For six of the seven experiments performed with diesel fuel, HNCO emissions were reduced by 5-40% with increasing $NH_3$ injections. For one of the diesel experiments and both the biodiesel experiments in which the measured emission factors (and measured concentrations) for HNCO were the lowest, HNCO emissions increased by 30-125% as the $NH_3$ injection was increased. In summary, it is unlikely that SCR-equipped engines running on diesel fuel are a source of HNCO even when the $NH_3$ injection exceeds stoichiometric rates. It is possible that the use of biodiesel reduces primary emissions of HNCO and that the HNCO enhancements in these experiments reflect slight contributions from the SCR chemistry that are undetectable during most of the diesel experiments. The HNCO response on SCR systems when using biodiesel needs to be explored in future work.".

4. I was intrigued that the modeled dry deposition velocity of HNCO was taken to be equal to HNO3. I think it would be helpful to state what the corresponding HNCO lifetime is in the model wrt/deposition and how much this assumption impacts model [HNCO]. It is easy to imagine a factor of 2 if not much more uncertainty in this assumption. It would be helpful to the reader to know how important this term is.

Since there were no direct laboratory or field measurements, we relied on two different nitrogen-based surrogates to model dry deposition of HNCO. The nitric acid ($HNO_3$) surrogate was used to simulate fast deposition while nitric oxide (NO) surrogate was used to simulate slow deposition. Nitric acid is expected to have a lifetime of a few hours while nitric oxide is expected to have a lifetime of a week or more with respect to dry deposition. Therefore, these species should capture the uncertainty inherent in modeling the HNCO dry deposition. Between the fast and slow surrogate simulations, HNCO concentrations increased by ~50% in urban areas and by a factor of 2 to 5 in rural/remote regions. On the one hand, these simulations show that HNCO is sensitive to model assumptions about dry deposition. On the other, however, a two orders of magnitude change in dry deposition rate only resulted in a 50% increase in HNCO concentrations. This is mostly likely because dispersion of the urban plume plays a much important role in controlling the HNCO concentration than deposition. We have added the following text as a separate paragraph in section 3.3: "Furthermore, we investigated the sensitivity of model predictions to dry deposition by using NO as the surrogate to model dry deposition of HNCO; NO has a much slower dry deposition lifetime (~weeks) than nitric acid (~hours) and we expected HNCO concentrations to increase. Model predictions suggested that HNCO concentrations increased by ~50% in urban areas and by a factor of 2 to 5 in rural/remote regions. This suggests that using nitric acid to model the dry deposition of HNCO could under-predict urban concentrations of HNCO by as much as 50%. It is worth noting that an order of magnitude change in the dry deposition lifetime only resulted in a 50% change in concentration, suggesting that dispersion, rather than deposition, plays an important role in controlling the urban HNCO concentrations.".

5. I understand that benzene and HNCO should be strongly correlated near the source region, but these two molecules have very different atmospheric lifetimes. It would be helpful for the authors to provide some comment on the limits of making such correlations for non-source regions.

We agree with the reviewer and have added the following sentence to the discussion around HNCO-benzene regressions: "we expect benzene and HNCO to correlate only in source and/or urban regions and the regression may not be applicable for remote/rural locations since HNCO and benzene may have very different atmospheric lifetimes".

**Reviewer 2**

This manuscript concerns the potential for diesel engines to be a source of isocyanic acid (HNCO), particularly when they are used with urea selective catalytic reduction (SCR) systems. The study is of relevance as HNCO has been mooted as a potential driver of negative health outcomes. The work is in two parts: 1) laboratory experiments to determine the emissions of HNCO from a test engine fitted with an SCR system, with different treatments of urea (the reducing agent), and running the engine at different loads; and 2) regional chemical transport model (CTM) simulations to probe the expected ambient HNCO concentrations arising from transport sources, given the emission factors determined in part 1 (over California).

For part 1, despite the potential for urea-SCR systems to produce HNCO, the authors report no enhancement of HNCO in the tailpipe emissions. The authors do find, however, a base level of HNCO emission from their system, as well as an emission of an HNCO photochemical precursor (likely an amide), whose emission they are able to estimate. From the emission

factors they determine, they estimate that ambient HNCO concentrations over California maximize at around 20-100 pptv (in the LA Basin), depending on assumptions related to deposition of HNCO. Overall, I think that this study is well done, well written, and a valuable addition to the growing HNCO literature, providing an impetus to others to investigate the sources, fate and impacts of this compound further. I would be happy to recommend publication of this study after consideration of my minor points below.

1. General: There are lots of long paragraphs throughout. Please consider breaking them up to make reading the study less daunting!
Wherever required, long paragraphs have been broken up into smaller paragraphs.

2. P2, L14: Stylistic note - have used "it is clear" in two consecutive sentences
We have removed the 'it is clear' from the final sentence.

3. P2, L37 (and throughout): Please take care that the reactions/reagents are formatted well (i.e. not italics)
The equations were added using equation editor in Microsoft Word and will likely be resolved during typesetting.

4. P3, L14: There are other limitations of the Young et al. study, such as assuming het chem is an irreversible sink. Consider bringing in the Barth et al. (2013) study at this point.
The paragraph the reviewer mentions is centered around a discussion of the treatment of sources and production pathways of HNCO. We do not think mentioning the Barth et al. (2013) study would be appropriate here. Instead, we have updated our paragraph on *Loss Mechanisms for HNCO* to discuss results from Barth et al. (2013) in relation to our work: "Roberts et al. (2011) have argued that the gas-phase reaction of HNCO with OH, heterogeneous reaction of HNCO on an aerosol surface, and photolysis of HNCO are too slow to be relevant in the atmosphere, and claimed that the only relevant loss mechanism for HNCO was dry and wet deposition. Young et al. (2012) investigated the influence of irreversible uptake of HNCO by clouds in a global model and found that this loss mechanism competed with dry deposition only when the cloud pH was 6 or higher. In a follow up study, Barth et al. (2013) used a detailed box model to suggest that the cloud uptake of HNCO was not irreversible (i.e., HNCO could be released into the gas-phase after the cloud evaporated) although HNCO concentrations could be significantly depleted if air parcels containing HNCO encountered low-level cumulus clouds. Barth et al. (2013) have also suggested that HNCO could be taken up by aqueous aerosols that might serve as a sink. In this work, we only modeled dry deposition of HNCO as a loss mechanism, ignore all other processes, and consequently provide an upper bound on our HNCO estimates. The dry deposition for HNCO is modeled assuming equivalence to nitric acid. We chose nitric acid since, like HNCO, nitric acid is extremely soluble in water at neutral pH. The HNCO precursor was modeled as NO to determine its dry deposition.".

5. P6, L1: Any comment on what your emissions might be if you scaled them with HCN? (Also, I believe Young et al. effectively scaled theirs with CO for non biomass burning emissions since the HCN emissions were scaled with CO).
The emissions processor for the UCD/CIT model and the version of the gas-phase chemical mechanism used in this work (SAPRC-11) do not model the emissions and chemistry of HCN and hence we would not be able to model the emissions of HNCO by scaling the HCN emissions.

6. P7, L19: "Comparison with earlier work" (no need for "inter"?)
We have changed the title.

7. P7, L23: Is "ensemble" needed?

We have edited the text to "We also compare our results to the average HNCO emissions from eight light-duty gasoline vehicles measured by Brady et al. (2014).".

8. P8, L15-: For the model sections... I appreciate that this is a basic model study, but it would be worth saying a bit more about the other uncertainties when modeling. E.g., uncertainties in emissions/chemistry of species important for OH, uncertainties in transport schemes, lack of (?) het chemistry, etc.

In this study, we have examined the uncertainty in the emissions and photochemical production of HNCO through the 'low' and 'high' simulations and investigated the uncertainty in dry deposition by using a fast ($HNO_3$) and slow (NO) surrogate. These uncertainties are discussed in Section 2.2. We did not model the heterogeneous chemistry of HNCO on aqueous aerosol nor did we model HNCO uptake by cloud water. Our model predictions hence provide an upper bound estimate of atmospheric HNCO. A brief discussion of the HNCO loss processes (aqueous aerosol and cloud water) is captured in the paragraph on *Loss mechanisms for HNCO*.

The UCD/CIT model uses meteorological inputs from WRFv3.4 that have been evaluated previously (Hu et al., 2015). Over the years, the UCD/CIT model has also been extensively evaluated against ambient measurements of gas- and particle-phase species in California (e.g., Hu et al., 2012; Jathar et al., 2015; Jathar et al., 2016) and found to be comparable in performance to other CTMs (e.g., Community Multiscale Air Quality model maintained by the EPA). Hence, we do not anticipate the HNCO model predictions to be very sensitive to transport schemes. The advantage of working with a model that has been evaluated earlier is that it has allowed us to focus on constraining the inputs and processes related to HNCO (see above paragraph). We have cited examples of recent efforts with this model in the section on *Chemical Transport Model*.

8. P8, L20: Some brief reminder of the context of the 1 ppbv value would be good here.

The following sentence has been added to the manuscript: "Roberts et al. (2011) argued that a 1 ppbv HNCO concentration would translate to a 100 µM aqueous HNCO concentration, which would be sufficient to result in carbamylation reactions that have been linked to adverse health outcomes".

9. P9, L3: Please explain the use of the benzene correlation

We had added the following sentence to the manuscript: "We chose benzene because Roberts et al. (2014) had developed an emissions ratios for HNCO with respect to benzene and Wentzell et al. (2013) had previously found ambient HNCO concentrations to vary linearly with benzene concentrations.".

10. P10, L1: Is agriculture also a source of HNCO precursors?

We have added 'agricultural burning' as a source of HNCO precursors.

11. P10, L4: "Using our experimentally-determined emission factors, we used a chemical..."

We have added the sentence as per the reviewer's suggestion.

12. P13, Fig 1 caption: "The fits, which parameterize..."

The sentence has been corrected.

13. P14, Fig 4 caption: Explain "low" and "high", say that Roberts et al. (2014) refers to CalNex (as per the panel), and please fix the color bars - how can LA be on the scale if the color bars stop at ~50% of the color bar value? (Should there be a triangle to indicate above 10 or 100? Or perhaps a note to say that the color bar saturates)

To address the 'low' and 'high comment, we have added the following sentence: "The low and high estimates used two different primary emissions and photochemical production parameterizations for diesel engines (see Table 1).". We have added the CalNex reference in the caption. We have added the following sentence to the caption with regards to the color bars: "The color scales in panels (a) and (b) have been truncated at 10 and 50 pptv respectively to visualize the statewide variability in HNCO concentrations. In both panels, the HNCO concentrations in the Los Angeles region exceed the concentrations captured on the color scale.".

**References**

Barth, M., Cochran, A., Fiddler, M., Roberts, J., and Bililign, S.: Numerical modeling of cloud chemistry effects on isocyanic acid (HNCO), Journal of Geophysical Research: Atmospheres, 118, 8688-8701, 2013.

Hu, J., Zhang, H., Ying, Q., Chen, S.-H., Vandenberghe, F., and Kleeman, M. J.: Long-term particulate matter modeling for health effect studies in California–Part 1: Model performance on temporal and spatial variations, Atmospheric Chemistry and Physics, 15, 3445-3461, 2015.

Hu, J., Howard, C. J., Mitloehner, F., Green, P. G., and Kleeman, M. J.: Mobile source and livestock feed contributions to regional ozone formation in Central California, Environ. Sci. Technol., 46, 2781-2789, 2012.

Jathar, S. H., Cappa, C. D., Wexler, A. S., Seinfeld, J. H., and Kleeman, M. J.: Multi-generational oxidation model to simulate secondary organic aerosol in a 3-D air quality model, Geosci. Model Dev., 8, 2553-2567, 10.5194/gmd-8-2553-2015, 2015.

Jathar, S. H., Mahmud, A., Barsanti, K. C., Asher, W. E., Pankow, J. F., and Kleeman, M. J.: Water uptake by organic aerosol and its influence on gas/particle partitioning of secondary organic aerosol in the United States, Atmospheric Environment, 129, 142-154, 2016.

Suarez-Bertoa, R., and Astorga, C.: Isocyanic acid and ammonia in vehicle emissions, Transportation Research Part D: Transport and Environment, 49, 259-270, 2016.